# Delta-Influence: Identifying Poisons via Influence Functions

**Wenjie Li**                      *liwj2022@shanghaitech.edu.cn*
*ShanghaiTech University*

**Jiawei Li**                      *li-jw19@mails.tsinghua.edu.cn*
*Tsinghua University*

**Pengcheng Zeng**                  *zengpch@shanghaitech.edu.cn*
*ShanghaiTech University*

**Christian Schroeder de Witt**           *christian.schroeder@eng.ox.ac.uk*
*University of Oxford*

**Ameya Prabhu**                  *ameya.prabhu@bethgelab.org*
*Tübingen AI Center*
*University of Tübingen*

**Amartya Sanyal**                  *amsa@di.ku.dk*
*University of Copenhagen*

**Reviewed on OpenReview:** *https://openreview.net/forum?id=4XtcG8NNaG*

## Abstract

Addressing data integrity challenges, such as unlearning the effects of targeted data poisoning after model training, is necessary for the reliable deployment of machine learning models. State-of-the-art influence functions, such as EK-FAC (Grosse et al., 2023) and TRAK (Park et al., 2023), often fail to accurately attribute abnormal model behavior to specific poisoned training data responsible for the data poisoning attack. In addition, traditional unlearning algorithms often struggle to effectively remove the influence of poisoned samples (Pawelczyk et al., 2024), particularly when only a few affected examples can be identified (Goel et al., 2024). To address these challenges, we introduce Δ-Influence, a novel approach that leverages influence functions to trace abnormal model behavior back to the responsible poisoned training data using just *one* poisoned test example, without assuming any prior knowledge of the attack. Δ-Influence applies data transformations that sever the link between poisoned training data and compromised test points without significantly affecting clean data. This allows detecting large negative shifts in influence scores following data transformations, a phenomenon we term as influence collapse, thereby accurately identifying poisoned training data. Unlearning this subset, e.g. through retraining, effectively eliminates the data poisoning. We validate our method across three vision-based poisoning attacks and three datasets, benchmarking against five detection algorithms and five unlearning strategies. We show that Δ-Influence consistently achieves the best unlearning across all settings, showing the promise of influence functions for corrective unlearning. Code is available at https://github.com/Ruby-a07/delta-influence.

## 1 Introduction

Machine learning models are increasingly deployed in critical sectors such as healthcare, finance, and autonomous systems (Davenport & Kalakota, 2019; Huang et al., 2020; Soori et al., 2023). This underscores the importance of ensuring model integrity and robustness against targeted data poisoning attacks. In such

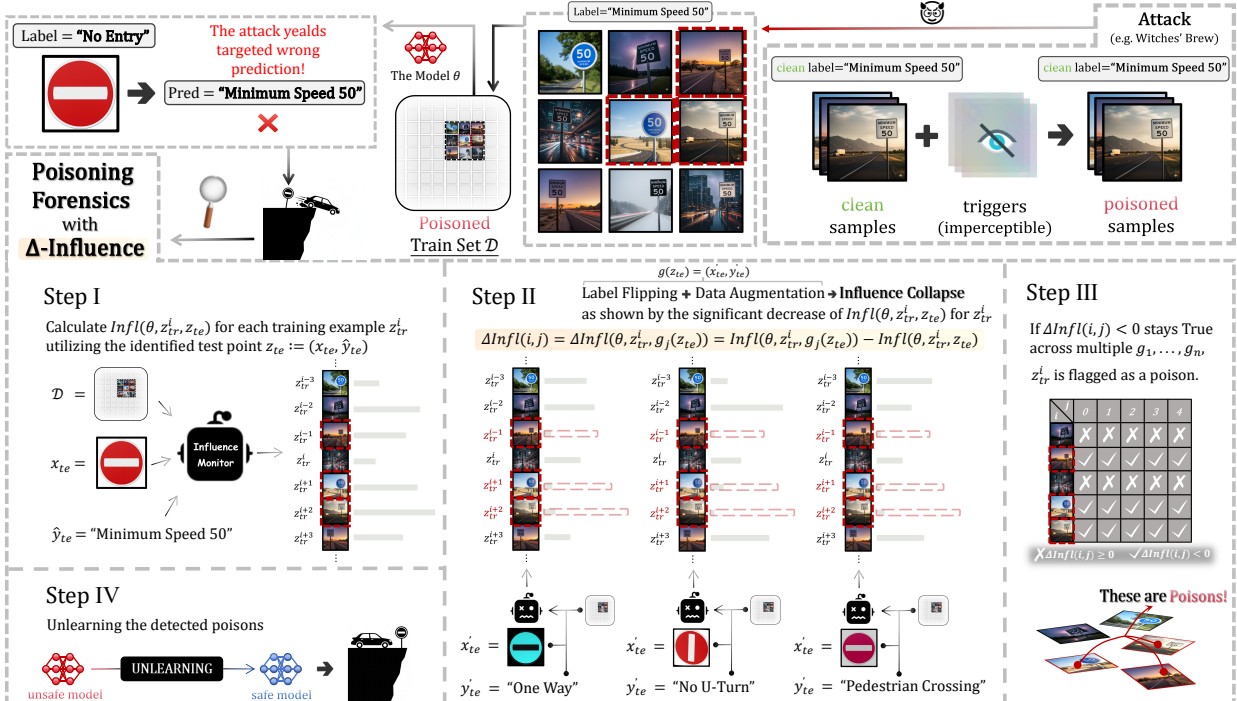

Figure 1: Our goal is to identify the training points responsible for the poisoning with an affected test point, so that retraining without these points can remove the attack from the model. State-of-the-art methods (Grosse et al., 2023; Park et al., 2023) detect only a few poisoned points with low precision, leaving the poisoning effect in the model and causing a large accuracy drop. $\Delta -$ Influence outperforms existing approaches by successfully recovering the clean model without sacrificing accuracy.

poisoning, adversaries intentionally manipulate training data by introducing carefully crafted, often imperceptible modifications (Chatila et al., 2021), leading to incorrect predictions or embedding specific malicious behaviors within the trained models (Fan et al., 2022). Given the large scale of modern datasets, identifying and removing all manipulated samples is typically impractical (Nguyen et al., 2024a; Goel et al., 2024). Therefore, a viable approach involves detecting and attributing the impact of data poisoning to a small set of *influential* training data points, which is *unlearned* to mitigate the data poisoning attack.

The challenge of effective unlearning largely depends on the extent of knowledge about the data poisoning attack. For example, Goel et al. (2024) demonstrate that retraining a model after removing a randomly sampled subset containing half of the manipulated data fails to eliminate poisoning in relatively simple attacks like BadNet (Gu et al., 2019). In contrast, retraining without the entire set of manipulated data successfully removes the attack. Furthermore, for more sophisticated poisoning strategies such as Witches' Brew (Geiping et al., 2021), Pawelczyk et al. (2024) reveal that existing unlearning algorithms are ineffective unless the model is retrained without the full manipulated set, even when full access to the manipulated data is available.

Building upon the framework of *Corrective Unlearning* introduced by Goel et al. (2024), our work addresses the setting in which the defender has identified a small set of affected test points. We note that detecting such affected data is a practical trigger for realizing that unlearning is necessary and thereby initiating the unlearning process, which can be regarded as a form of *poisoning forensics*: starting from a compromised output, we trace back to the culpable training examples whose removal neutralizes the attack. In practice, such "perpetrators" typically surface through (i) deployment observation of anomalous behavior (*e.g.*, a permission system granting administrative access to an unknown user, a stop sign being misclassified as a minimum speed-limit sign) or (ii) deliberate in-house stress testing (*e.g.*, red-teaming, white-hat). A key advantage of our method is that it requires only the logically unavoidable minimum that at least *one affected*

*test point can be identified*; other methods, while effective in their respective settings, typically assume a larger identified set (Min et al., 2025; Coalson et al., 2025). Leveraging this poisoned test point, we propose a *detect-then-unlearn* pipeline: first, identifying a critical set of manipulated training points responsible for the compromised prediction; and second, applying unlearning algorithms to remove the influence of these points from the model. Our approach departs from prior unlearning works that often presuppose the availability of a "forget set", a subset of known poisoned training points (Goel et al., 2023; Kurmanji et al., 2023; Foster et al., 2024). We argue that such availability can be challenging to satisfy in practice, especially in complex clean-label attacks like Witches' Brew (as illustrated in the 'Attack' panel of Figure 1).

Within this framework, *influence functions* (Koh & Liang, 2017) serve as a natural tool for attributing model predictions to specific training data points. However, recent studies (Grosse et al., 2023; Nguyen et al., 2024b; Bae et al., 2024) have indicated that state-of-the-art influence functions struggle to accurately identify the manipulated data when used in a naive manner. Our experiments in Section 3 also corroborate this finding. To address this, we introduce $\Delta-$Influence, an approach that enhances influence functions to reliably identify a critical set of training data points necessary for unlearning data poisoning without compromising model accuracy. Instead of directly calculating each training point's influence on a poisoned test point, $\Delta-$Influence assesses the change in influence scores before and after perturbing the test point through (i) label flipping and (ii) image transformation. As ablation studies in Section 4 show, label flipping is essential for breaking the association between poisoned data and the affected test point, while image transformations introduces randomness that reduces false positive rates by preserving the influence of benign data.

To assess the effectiveness of $\Delta-$Influence and the broader applicability of influence functions in this context, we apply our method to three prominent targeted data poisoning attacks: Frequency Trigger (Zeng et al., 2021), Witches' Brew (Geiping et al., 2021), and BadNet (Gu et al., 2019). We compare our approach against multiple defenses (Chen et al., 2018; Tran et al., 2018; Zeng et al., 2021; Grosse et al., 2023; Park et al., 2023) that operate with similar or less information about the poisoning than $\Delta-$Influence. Each attack presents unique challenges for detection and mitigation, as evidenced by the varying performance of existing detection methods across different attacks. Additionally, we conduct experiments using known unlearning algorithms to unlearn the poisoning attack using the identified set. These experiments provide a comprehensive comparison of these unlearning algorithms. For example, the gradient ascent-based method SCRUB (Kurmanji et al., 2023) can successfully unlearn some poisoning attacks (*e.g.* BadNet) when the detected set of training poisons is reasonably accurate. However, its resultant accuracy drops significantly if the detected set includes many falsely flagged clean examples. In contrast, methods like EU and CF (Goel et al., 2024) are surprisingly robust to false positives, delivering the best unlearning and accuracy. Overall, our experiments demonstrate that $\Delta-$Influence consistently outperforms existing algorithms across all settings, offering a robust defense against sophisticated data poisoning attacks while preserving accuracy.

## 2 Using Influence functions to detect poisons

In this section, we present how influence functions can be leveraged to unlearn data poisoning attacks and introduce our primary algorithm, $\Delta-$Influence.

Consider an example where an adversary modifies a subset of training images belonging to a specific *victim* class by adding a subtle trigger and altering their labels to a *target* class. These manipulated examples, referred to as *poisons*, are incorporated into the training dataset. Consequently, the trained model learns to misclassify any test image from the victim class containing the trigger as belonging to the target class, while maintaining normal predictions on other test images.

Influence functions (Koh & Liang, 2017) provide a mechanism to quantify the contribution of each training example to a particular prediction. By computing the influence of each training point on the prediction of the selected test point, we can identify the most influential training samples responsible for abnormal behavior. Specifically, poisoned examples typically exert a significant influence on the affected test predictions; this makes it possible to distinguish the poisons through their influence scores. Thus, influence functions offer a natural algorithm for tracing abnormal predictions back to responsible poisoned training data.

However, our experiments in Section 3, along with several recent studies (Nguyen et al., 2024b; Bae et al., 2024; Li et al., 2024b), demonstrate that naively applying state-of-the-art influence functions fails to accurately identify poisoned points. This limitation necessitates the development of a more robust method to effectively utilize influence functions for detecting and unlearning data poisoning.

## 2.1 Our Algorithm: $\Delta$-Influence

To address the shortcomings of the naive approach, we introduce $\Delta - \text{Influence}$. The core idea is to monitor the changes in influence scores of training data points when the affected test point undergoes various transformations.

**Notations.** Let $z_{\text{tr}}^i \coloneqq \left(x_{\text{tr}}^i, y_{\text{tr}}^i\right)$ denote a labeled training data point, where $x_{\text{tr}}^i \in \mathcal{X}$ represents the $i_{th}$ training input (e.g., an image for vision tasks) and $y_{\text{tr}}^i \in \mathcal{Y}$ represents the label. Let $\theta^\star$ represent the trained model parameters optimized on the training dataset. For a given test point $z_{\text{te}} \coloneqq (x_{\text{te}}, \hat{y}_{\text{te}})$ with predicted label $\hat{y}_{\text{te}}$, the influence function quantifying the impact of $z_{\text{tr}}^i$ on the loss of $z_{\text{te}}$ is:

$$\text{Infl}\left(\theta^\star, z_{\text{tr}}^i, z_{\text{te}}\right) = \nabla_\theta \mathcal{L}\left(z_{\text{te}}, \theta^\star\right)^\top \mathbf{H}^{-1} \nabla_\theta \mathcal{L}\left(z_{\text{tr}}^i, \theta^\star\right), \tag{1}$$

where $\mathcal{L}(z, \theta^\star)$ is the loss evaluated at the point $z$ with parameters $\theta^\star$ and $\mathbf{H}$ is the Hessian of the loss function with respect to $\theta$ at $\theta^\star$. Higher influence values indicate a greater contribution of the training point $z_{\text{tr}}^i$ to the prediction on the test point $z_{\text{te}}$.

**Monitoring Change in Influence.** Our goal is to attribute the predicted label $\hat{y}_{\text{te}}$ of a poisoned test point $z_{\text{te}}$ to a subset of training points $\mathcal{P} = \{z_{\text{tr}}^1, \ldots, z_{\text{tr}}^k\}$ responsible for the misclassification. To achieve this, we monitor the change in influence scores $\text{Infl}\left(\theta, z_{\text{tr}}^i, z_{\text{te}}\right)$ for each training data point $z_{\text{tr}}^i$ when the test point $z_{\text{te}}$ undergoes a set of transformations.

Formally, let $g_j$ be a transformation applied to the test point $z_{\text{te}} = (x_{\text{te}}, y_{\text{te}})$, consisting of pairing the test image with a random label $y_{\text{te}}'$ and applying standard data augmentations such as blurring, color jitter and rotating to $x_{\text{te}}$ (see Appendix B.2 for the list of all transformations). Note that we utilize common data augmentation techniques without designing any poison-specific transformations, suggesting the broad applicability of $\Delta$-Influence. We consider such simplicity to be a key strength of our contribution. Then, for each transformation $g_j$, we compute the change in influence score as

$$\Delta\text{Infl}(\theta, z_{\text{tr}}{}^i, g_j\left(z_{\text{te}}\right)) = \text{Infl}\left(\theta, z_{\text{tr}}{}^i, g_j(z_{\text{te}})\right) - \text{Infl}\left(\theta, z_{\text{tr}}{}^i, z_{\text{te}}\right). \tag{2}$$

For brevity, we denote this change as $\Delta\text{Infl}(i, j)$, where $i$ and $j$ index the training point and the transformation function, respectively.

**Influence Collapse.** Computing the $\Delta - \text{Influence}$ is motivated by the following two observations, which we refer to as *Influence Collapse*. Let $z_{\text{te}}$ be the affected test point.

1. **Negative Change for Poisons:** For all manipulated training samples $z_{\text{tr}}^i \in \mathcal{P}$ and transformations $g_j$, the change in influence $\Delta\text{Infl}(i, j)$ is consistently negative.

2. **Minimal Change for Clean:** For all clean training samples $z_{\text{tr}}^k \notin \mathcal{P}$ and transformations $g_j$, the change in influence $\Delta\text{Infl}(k, j)$ is significantly smaller in magnitude and often positive in value, for most transformations.

This is illustrated in Figure 2, where $\Delta\text{Infl}(i, j)$ is consistently negative for poisoned samples across all transformations, whereas it often remains near zero (compared to that of poisons) or shows no clear trend for clean examples. However, Figure 2 shows that this is not consistently the case for all clean examples (with some values being considerably small), which brings us to the next component.

**Boosting Using Multiple Transformations.** The above discussion shows that the change in $\Delta\text{Infl}(i, j)$ can be used as a score function for detecting whether $z_{\text{tr}}^i$ is manipulated. However, this score function is

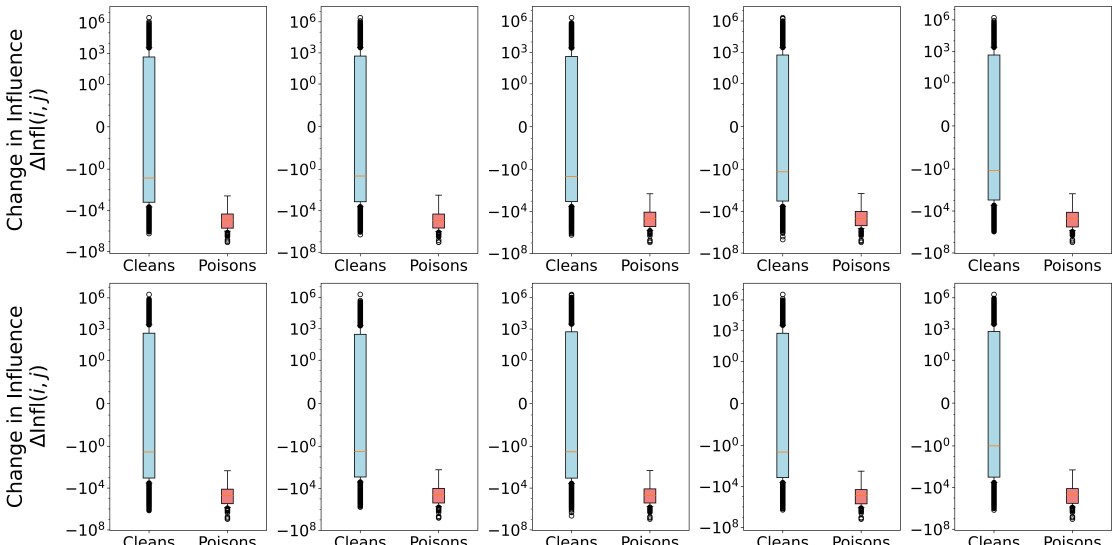

Figure 2: The Influence Score Change ($\Delta\text{Infl}(i, j)$) for 125 poisoned training points (orange) and 49,875 clean training points (light blue) on the Frequency Trigger with CIFAR100. Each plot shows the score change for a different transformation applied to the affected test image. Results show a consistent drop in scores for all poisoned examples while clean examples exhibit no clear trend.

a relatively weak classifier, especially for clean points, as seen in Figure 2. To overcome this problem, we use classical ideas from bagging and apply multiple random transformations $g_1, \ldots, g_{n_b}$ to obtain a series of weak classifiers, where each classifier flags the example if its corresponding score is sufficiently negative. We use $n_b$ transformations to obtain $n_b$ weak classifiers.

Then, we combine the classifiers using a count-based decision rule. Specifically, we flag $z^i_{\text{tr}}$ as manipulated if a sufficiently large number of weak classifiers also flag it. Note that this happens if a large number of transformations simultaneously lead to a negative change in influence score for the example. The key hypothesis we leverage here is that for most clean points, a few transformations will always result in a positive change in influence $\Delta\text{Infl}(i, j)$.

**Unlearning identified points.** Once the set of poisoned training points $\mathcal{P}$ is identified using $\Delta-\text{Influence}$, the next step is to unlearn them to mitigate the data poisoning attack. We employ several unlearning algorithms (Goel et al., 2023; Kurmanji et al., 2023; Golatkar et al., 2020; Foster et al., 2024) to remove the influence of $\mathcal{P}$ from the trained model $\theta^\star$. In practice, the choice of unlearning algorithm may depend on factors such as computational efficiency, scalability, and the specific characteristics of the poisoning attack. In this work, we look at several popular unlearning algorithms including retraining from scratch (denoted as EU (Goel et al., 2023)), CF (Goel et al., 2023), SSD (Foster et al., 2024), SCRUB (Golatkar et al., 2020), and BadT (Kurmanji et al., 2023).

## 2.2 Full Algorithm

To summarise, the full pipeline of detection and unlearning in $\Delta - \text{Influence}$ proceeds as follows:

1. Initialization Begin with trained model $\theta^\star$, a poisoned test point $z_{\text{te}}$, and the entire training dataset $\mathcal{D} = \{z^i_{\text{tr}}\}^N_{i=1}$.

2. Transformations Apply a diverse set of transformations $\mathcal{G} = \{g_j\}^{n_b}_{j=1}$ to the poisoned test point $z_{\text{te}}$ to obtain multiple $z'_{\text{te}} = g_j(z_{\text{te}})$.

3. Influence Score For each training data point $z_{\mathrm{tr}}^i \in \mathcal{D}$ and each transformation $g_j \in \mathcal{G}$, compute the change in influence score $\Delta\mathrm{Infl}(i,j)$ as defined in Equation (2).

4. Boosting and Detection For each training data point $z_{\mathrm{tr}}^i$, aggregate the influence score changes across all transformations. If the number of significant negative changes exceeds $n_b - n_{\mathrm{tol}}$, flag $z_{\mathrm{tr}}^i$ as a poisoned sample ($n_{\mathrm{tol}}$ is a hyperparameter, see Appendix B.4). If the number of negative changes exceeds a predefined threshold $\tau$, flag $z_{\mathrm{tr}}^i$ as a poisoned sample.

5. Unlearning Once the set of poisoned training points $\mathcal{P}$ is identified, apply unlearning algorithms to remove their influence from the trained model $\theta^\star$.

In the next section, we test and evaluate the above algorithm on several datasets, data poisons, and unlearning algorithms to compare it with existing approaches.

## 3 Experiments

### 3.1 Experimental Setup

**Attacks.** Ensuring broad coverage and robustness, we evaluate against three distinct types of attacks:

1. Frequency Trigger (Zeng et al., 2021): In this approach, along with changing the label, a trained, imperceptible pattern is embedded in both the spatial and frequency domains, thereby encompassing the whole image. As shown in Alex et al. (2024), these patterns are difficult to detect by both human and automated methods, making the poisoned samples challenging to identify.

2. Clean Label Attack (Witches' Brew) (Geiping et al., 2021): Unlike Frequency Trigger, this attack adds an imperceptible pattern to images without altering their labels. The poisoned samples appear benign since their labels are consistent with their content, yet they cause the model to learn incorrect associations, leading to misclassifications during inference. As shown in Pawelczyk et al. (2024), these patterns are difficult to unlearn using unlearning algorithms.

3. Patch Trigger (BadNet) (Gu et al., 2019): Also studied in Goel et al. (2024), this attack involves adding a subtle patch to the corner of selected training images and altering their labels to a designated target class. The presence of the patch causes the model to misclassify any test image containing the patch into the target class while maintaining normal performance on other inputs.

**Model and Datasets.** We utilize the CIFAR10 and CIFAR100 datasets (Krizhevsky, 2009) and a ResNet18 model (He et al., 2015), following the standard benchmarks and models used in the state-of-the-art machine unlearning setup (Pawelczyk et al., 2024). For CIFAR10, we poison 500 training images (1% of the dataset), while for CIFAR100, we poison 125 training images (0.25% of the dataset) for all attack types except BadNet, which requires a higher size of 350 samples to be effective. The victim class and attack class (when different) are selected randomly. Detection methods are tuned on a small validation set using cross-validation techniques. Hyperparameters such as threshold values and clustering parameters are optimized based on validation performance metrics to achieve the best balance between detection accuracy and false positive rates. Detailed hyperparameter settings and our code are provided in the Appendix B to ensure reproducibility.

**Compared Methods.** We compare the detection performance of existing popular methods in the data poisoning literature by adapting them to our setting. Additionally, we include the state-of-the-art methods for computing influence function: EK-FAC (Grosse et al., 2023) and TRAK (Park et al., 2023) as baselines. Our $\Delta - \mathrm{Influence}$ method is built upon EK-FAC.

1. Activation Clustering-Based Detection (Chen et al., 2018) identifies backdoored samples by clustering the activations of the last hidden layer for each class. If a class's activations can be effectively clustered into two distinct groups, the smaller cluster is deemed to contain poisoned samples and is subsequently removed for retraining.

2. Spectral Signature-Based Detection (Tran et al., 2018) employs singular value decomposition on the activations of the last hidden layer per class. Samples with high values in the first singular dimension are flagged as poisoned and removed based on a predefined hyperparameter threshold.

3. Frequency-Based Detection (Zeng et al., 2021) performs frequency analysis by building a classifier on the discrete cosine transforms of synthetic images containing hardcoded backdoor-like features. It identifies poisoned examples by detecting these frequency-based patterns.

4. EK-FAC (Grosse et al., 2023) serves as our baseline method for using influence functions in poison detection. It calculates influence scores for every training sample based on one known affected test sample. Samples with average scores exceeding a predefined threshold are removed.

5. TRAK (Park et al., 2023) uses another popular implementation of influence functions when thresholding.

**Metrics.** We evaluate our algorithm using four key metrics. All metrics are averaged over three random seeds.

1. True Positive Rate (TPR): Fraction of identified poisoned samples out of the total poisoned samples in train set.
$$\frac{\text{Number of correctly flagged poisoned samples}}{\text{Total number of poisoned samples}} \times 100\%$$

2. Precision: Proportion of correctly identified poisoned samples among all flagged samples. It captures the trade-off between detection accuracy and model utility.
$$\frac{\text{Number of correctly flagged poisoned samples}}{\text{Total number of samples flagged as poisoned}} \times 100\%$$

3. Poison Success Rate (PSR): Fraction of poisoned test samples that are misclassified into the target (incorrect) class.
$$\frac{\text{Number of poisoned samples classified as target}}{\text{Total number of poisoned samples}} \times 100\%$$

4. Test Accuracy: The performance on unpoisoned test samples, measuring drop in model utility.
$$\frac{\text{Number of correct predictions on test set}}{\text{Total number of test samples}} \times 100\%$$

5. Area under the ROC curve (AuROC): Trade-off between TPR and Precision due to the choice of threshold.

### 3.2 Main Results

We present our experimental findings across the above metrics and compare the performance of $\Delta-$ Influence against several baselines. Specifically, we report the precision, TPR, and AuROC of detecting poisons in Table 1, and the overall PSR and test accuracy after retraining without the identified set in Figure 3.

**Performance of $\Delta$-Influence.** As illustrated in Figure 3, $\Delta-$ Influence consistently achieves a poison success rate below 5% across all three types of poisoning attacks and both datasets. This success rate is marked by a ✓, while unsuccessful detections are marked by a ×. In contrast, the next best methods, Activation Clustering (ActClust) and EK-FAC, succeed in only 3 out of 6 cases, as highlighted in Table 1. This showcases the substantial improvement in performance gained by $\Delta-$ Influence.

Among the baseline methods, EK-FAC outperforms ActClust by minimizing the drop in test accuracy, also indicated by a higher precision in Table 1. Furthermore, $\Delta$-Influence consistently achieves the highest precision, offering better performance with minimal accuracy loss compared to the other methods. Additional experiments detailed in Section 4.1 demonstrate that both label and input augmentations are necessary for $\Delta-$ Influence.

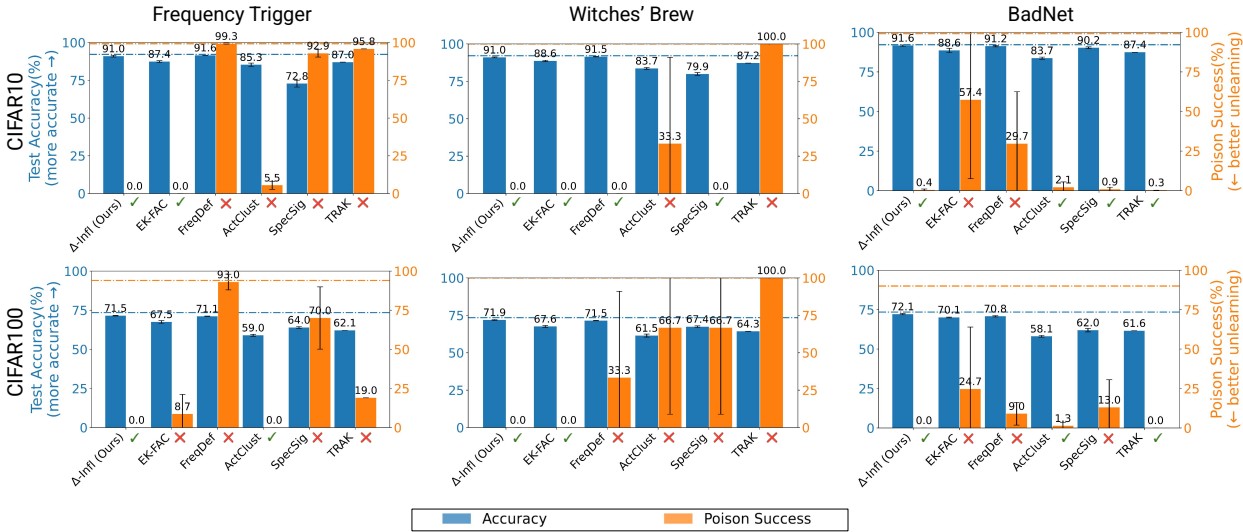

Figure 3: Poison Success Rate and Test Accuracy. This table shows both poison unlearning effectiveness and model utility. A method is considered successful if the PSR is below 5%, marked by ✓, with unsuccessful methods marked by ×. Δ-Influence is successful in 6/6 cases, while the closest competitors succeed in only 3/6. Additionally, Δ-Influence nearly perfectly preserves test accuracy. Figure structure from (Pawelczyk et al., 2024).

**Variance across Poisons.** Our analysis shows that the BadNet poison can be effectively removed without identifying all poisoned samples, reaffirming that it is realatively easy to eliminate. Based on these results, we advocate that the corrective unlearning literature should benchmark proposed algorithms on the more challenging frequency-based poisons (Zeng et al., 2021), which require detecting nearly all poisoned samples and are notably harder to remove with a partial subset. This was also identified to be difficult in previous work (Alex et al., 2024).

Surprisingly, in the case of the Witches' Brew attack on CIFAR-10, our Δ−Influence method often identifies fewer but a sufficient number of true poisoned samples compared to other methods. We attribute this to the unique behavior of this particular poison. Δ−Influence effectively identifies the samples most responsible for the misclassification, and in Witches' Brew, only a few samples are truly effective for poisoning. Additional experiments in Section 4.3 show that removing the complement of detected poisons does not allow the model to recover, despite the complement set being similar in size or larger.

**Conclusion.** Overall, Δ−Influence offers an effective mechanism for unlearning data poisoning attacks without significantly impacting model performance. Crucially, it requires *no prior knowledge of the attack method*, making it more generalizable across various poisoning strategies.

# 4 Unpacking Key Factors in Δ-Influence

We present a series of additional analyses designed to improve the understanding of Δ−Influence. Specifically, we explore: (i) individual contributions of image and label perturbations, (ii) effectiveness of various unlearning algorithms, (iii) a counterfactual analysis to determine whether the detected samples are solely responsible for enabling poisoning in the Witches' Brew attack, and (iv) the unreliability of using a known training poison as an attribution target (see Appendix K).

## 4.1 Perturbing Only Images or Labels

**Setup.** To distinguish the contributions of image and label perturbations in our Δ−Influence method, we conduct an ablation study by evaluating the two key components separately:

Table 1: Comparison of Precision & TPR & AuROC across methods and dataset for detecting poisoned samples. Green indicates successful unlearning (PSR ≤ 5%, while red indicates failed unlearning (see fig. 3 for exact poisoning success rates). We evaluate the precision and TPR of detecting poisoned training samples. SpecSig Tran et al. (2018), ActClust Chen et al. (2018), TRAK Park et al. (2023) and EK-FAC Grosse et al. (2023) yield low precision, flagging many clean samples as poisoned. FreqDef Zeng et al. (2021) and $\Delta-$Influence better preserve clean data, though FreqDef shows a significantly lower TPR, missing many true poisoned samples. For BadNet, the poisoning success rate correlates with the number of detected poisoned samples, making the attack in Goel et al. (2024) relatively easy to unlearn. In contrast, the Frequency attack requires nearly all poisoned samples to be removed for recovery, making it particularly challenging. Surprisingly, the Witches' Brew setting is easier than anticipated Pawelczyk et al. (2024), requiring only a few key samples—mainly identified by influence functions—to be removed for effective unlearning.

| Method | Metric | CIFAR10 | | | CIFAR100 | | |
|---|---|---|---|---|---|---|---|
| | | Frequency Trigger | Witches' Brew | BadNet | Frequency Trigger | Witches' Brew | BadNet |
| SpecSig | Precision | 1.3% | 1.4% | 3.6% | 0.5% | 0.3% | 1.3% |
| | TPR | 88.3% | 96.8% | 88.3% | 78.4% | 35.2% | 82.6% |
| | AuROC | 0.53 | 0.88 | 0.82 | 0.72 | 0.58 | 0.76 |
| ActClust | Precision | 2.2% | 2.1% | 2.2% | 0.6% | 0.3% | 1.6% |
| | TPR | 99.1% | 93.4% | 94.9% | 100% | 55.2% | 96.3% |
| | AuROC | 0.77 | 0.75 | 0.79 | 0.79 | 0.53 | 0.78 |
| FreqDef | Precision | 0.4% | 10.2% | 8.0% | 0.1% | 1.8% | 5.3% |
| | TPR | 3.2% | 93.6% | 72.3% | 2.4% | 78.4% | 85.7% |
| | AuROC | 0.32 | 0.98 | 0.97 | 0.28 | 0.91 | 0.97 |
| TRAK | Precision | 1.4% | 1.0% | 1.9% | 0.5% | 0.2% | 1.4% |
| | TPR | 70.6% | 49.8% | 93.6% | 96.8% | 48.0% | 100% |
| | AuROC | 0.73 | 0.50 | 0.60 | 0.79 | 0.49 | 0.58 |
| EK-FAC | Precision | 2.9% | 0.8% | 2.8% | 0.9% | 0.4% | 3.2% |
| | TPR | 100% | 17.4% | 67.1% | 96.8% | 47.2% | 70.0% |
| | AuROC | 0.89 | 0.57 | 0.87 | 0.94 | 0.68 | 0.71 |
| $\Delta$-Infl (Ours) | Precision | 13.3% | 3.3% | 17.6% | 2.9% | 2.1% | 37.3% |
| | TPR | 100% | 19.4% | 99.1% | 100% | 62.4% | 96.9% |
| | AuROC | 0.96 | 0.38 | 0.95 | 0.96 | 0.75 | 0.82 |

Table 2: Comparison of Precision & TPR & AuROC across Label-Only, Image-Only and combined transformation of affected image. Green indicates successful unlearning (PSR < 5%), while red indicates unsuccessful unlearning (See Appendix for exact PSR). Label-only augmentations are highly effective in detecting poisoned samples, whereas image-only augmentations perform poorly. Conversely, image-only augmentations significantly reduce the FPR, preserving more clean data and improving detection precision.

| Method | Metric | CIFAR10 | | | CIFAR100 | | |
|---|---|---|---|---|---|---|---|
| | | Frequency Trigger | Witches' Brew | BadNet | Frequency Trigger | Witches' Brew | BadNet |
| Ours (Label-Only) | Precision | 6.3% | 1.2% | 4.0% | 1.1% | 0.8% | 3.1% |
| | TPR | 100% | 24.2% | 97.5% | 100% | 73.6% | 99.1% |
| | AuROC | 0.94 | 0.48 | 0.90 | 0.92 | 0.69 | 0.79 |
| Ours (Img-Only) | Precision | 28.9% | 2.7% | 14.4% | 0.6% | 0.3% | 7.6% |
| | TPR | 26.4% | 13.2% | 68.9% | 62.4% | 40.8% | 50.6% |
| | AuROC | 0.51 | 0.25 | 0.85 | 0.78 | 0.62 | 0.76 |
| Ours (Both) | Precision | 13.3% | 3.3% | 17.6% | 2.9% | 2.1% | 37.3% |
| | TPR | 100% | 19.4% | 99.1% | 100% | 62.4% | 96.9% |
| | AuROC | 0.96 | 0.38 | 0.95 | 0.96 | 0.75 | 0.82 |

1. Modify Label ($\Delta-$ Influence (Label-Only)): Conversely, in this baseline, we only modify the test point's labels while keeping the images unchanged. This setup helps evaluate the effect of label manipulation on detecting the influence of poisoned training points.

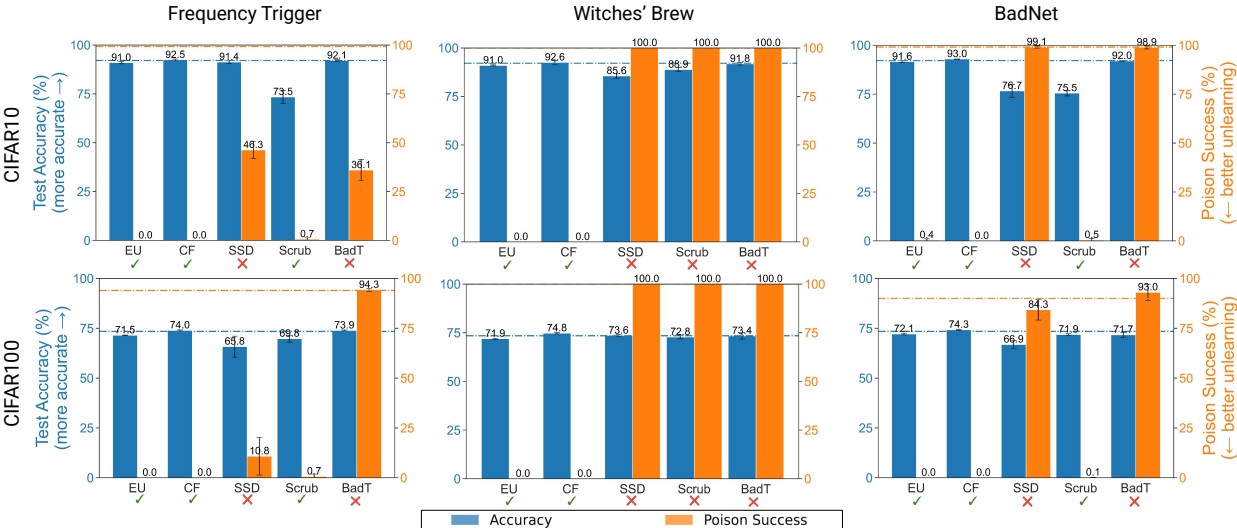

Figure 4: Poison Success Rate and Test Accuracy for Unlearning Methods Applied on Samples Identified by $\Delta-$Influence. Catastrophic Forgetting (CF) and Exact Unlearning (EU) from Goel et al. (2023) perform best, effectively unlearning poisoned samples while maintaining test accuracy. In contrast, SSD (Foster et al., 2024) and SCRUB (Kurmanji et al., 2023) struggle with false negatives, leading to significant accuracy drops, while BadT (Chundawat et al., 2023) fails to unlearn effectively. We recommend EU or CF as strong baselines and highlight the need for future methods to improve robustness against false positives.

2. Modify Image ($\Delta-$Influence (Img-Only)): In this baseline, we exclusively modify the test images without altering their labels. This allows us to isolate the impact of image transformations on the model's ability to detect poisoned data.

Both ablations are benchmarked across the same datasets and poisoning attacks, utilising identical metrics to ensure consistency in evaluation. The goal is to understand the individual and combined effects of image and label perturbations on the detection performance of $\Delta-$Influence.

**Results.** As depicted in Table 2, our ablation study reveals that label-only augmentations achieve high TPR across all poisoning types and datasets, effectively identifying almost all poisoned samples. However, this leads to low precision, resulting in the unnecessary removal of a significant number of clean samples. On the other hand, image-only augmentations exhibit poor TPR, failing at the core task but also rejects less clean samples (higher precision). In contrast, $\Delta-$Influence leverages both label and image perturbations to achieve a balanced performance and detects sufficient key poisoned samples while rejecting lesser clean samples (see Figure 6 in Appendix for detailed unlearning performance). Our ablation study underscores the necessity of incorporating both label and image augmentations in the $\Delta-$Influence method. Label flippings are pivotal for enhancing detection accuracy, while image transformations play a critical role in minimizing false positives.

**Conclusion.** Our ablation study underscores the necessity of incorporating both label and image augmentations in the $\Delta-$Influence method. Label perturbations are pivotal for enhancing detection accuracy, while image augmentations play a critical role in minimizing false positives.

## 4.2 Which Unlearning Methods Work?

**Setup.** To evaluate the effectiveness of various unlearning algorithms when paired with our $\Delta-$Influence method, we fix the influence scoring method to $\Delta-$Influence and vary the unlearning algorithm. We benchmark several corrective unlearning approaches, including exact unlearning methods such as EU (Goel et al., 2023), CF (Goel et al., 2023), as well as approximate unlearning methods such as SSD (Foster et al., 2024), SCRUB (Golatkar et al., 2020), and BadT (Kurmanji et al., 2023). Following (Pawel-

czyk et al., 2024), for approximate unlearning methods we reduce the compute budget to 10% of the original training budget (*e.g.*, 4 unlearning epochs when the original model is trained for 40 epochs). Meanwhile, to avoid confounding factors (*i.e.*, whether failures are due to poor detection or to limitations of imperfect approximate unlearning), we allocate the full training budget to exact unlearning methods EU and CF so that they serve as gold-standard baselines. All methods are implemented using the codebase and training protocols from Goel et al. (2024). Further implementation details are provided in Appendix B.

**Results.** As illustrated in Figure 4, our evaluation reveals that CF performs comparably to EU, achieving near-perfect poison removal. Both CF and EU remain robust against false positives, maintaining high test accuracy. In contrast, approximate unlearning methods are less reliable overall. While SCRUB can successfully remove poisons in some cases (Frequency Trigger and BadNet), it does so at the expense of model utility due to its susceptibility to false positives. BadT and SSD completely fail to unlearn poisons effectively.

**Conclusion.** We recommend EU or CF as competitive baselines for corrective unlearning using influence functions, and also highlight the importance of robustness towards false positives.

### 4.3 Counterfactual Analysis: Do Detected Samples Account for Poisoning in Witches' Brew?

Table 3: **Does the Detected Set Truly Influence the Poison?** For Witches' Brew, we test the "Original" set, representing the poisoned samples identified by $\Delta -$ Influence, and the "Complement" set, which includes all other poisoned samples not detected. The absence of a drop in poison success rate when removing the complement set suggests that the detected set fully captures the poisoning effect. Conversely, removing the detected set completely eliminates the poisoning effect.

| $\Delta$-Influence Set | TPR($\uparrow$) | Poison Success Rate ($\downarrow$) | Test Accuracy ($\uparrow$) |
|---|---|---|---|
| **CIFAR10** | | | |
| Original | 19.4% | 0% | 91.0% |
| Complement | 80.6% | 100% | 92.2% |
| **CIFAR100** | | | |
| Original | 62.4% | 0% | 71.9% |
| Complement Set | 37.6% | 100% | 72.8% |

**Setup.** The analysis compares the original detected set of poisoned samples in Witches' Brew to its complement set (i.e., all poisoned samples except those detected by $\Delta -$ Influence). This aims to assess whether the detected set exclusively accounts for the poisoning effect.

**Results.** As presented in Table 3, the removal of the "Original" detected set (19.4% TPR for CIFAR10 and 62.4% TPR for CIFAR100) results in 0% poison success rate, effectively unlearning the poisoning. In stark contrast, removing the "Complement" set (80.6% TPR for CIFAR10 and 37.6% TPR for CIFAR100) maintains a poison success rate of 100%, indicating that the undetected samples do not sufficiently contribute to the poisoning. The complement set achieves higher test accuracy simply because it only contains unaffected samples without false positives. These results demonstrate that our detected subset accounts for nearly all the poisoning effects in Witches' Brew, highlighting the unusual nature of this particular poison as well as the precision of our $\Delta -$ Influence algorithm.

**Conclusion.** These results demonstrate that our detected subset accounts for nearly all the poisoning effects in Witches' Brew, highlighting the unusual nature of this particular poison as well as the precision of our $\Delta -$ Influence algorithm.

### 4.4 Scaling Results to ImageNette

**Setup.** We evaluate the scalability and consistency of $\Delta -$ Influence on a more complex dataset, Imagenette. The setup is consistent with the experiments in Section 3 with specific adjustments (details in Appendix D.2).

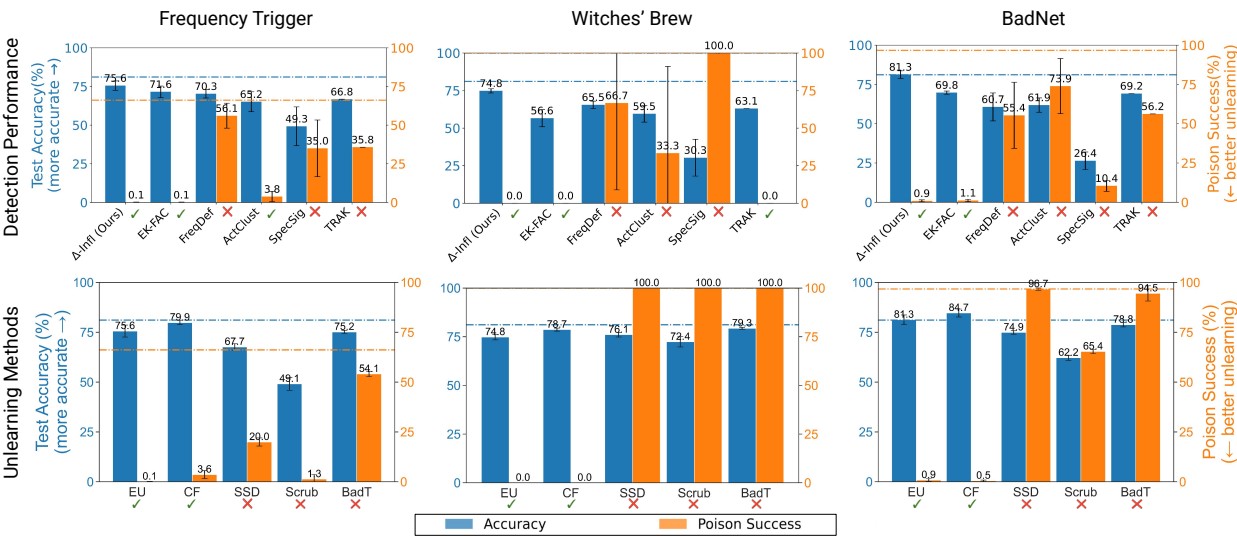

Figure 5: **Scaling to Imagenette.** Results on Imagenette are consistent with previous findings.

**Results.** Replicating our prior conclusions on Imagenette, Figure 5 illustrates that $\Delta$ − Influence continues to achieve the most effective poison unlearning across all attack types, maintaining minimal accuracy loss. Notably, the EK-FAC baseline also successfully unlearns all poisons but incurs a higher false positive rate, leading to significant drops in test accuracy due to the unnecessary removal of clean samples. When applying various unlearning algorithms to the samples identified by $\Delta$ − Influence, both CF and EU perform consistently well with CF achieving notably higher accuracy during poison unlearning compared to EU. In contrast, approximate unlearning methods perform substantially worse.

## 5 Conclusion

We address a critical issue in corrective machine unlearning: identifying training samples whose removal can unlearn a data poisoning attack. $\Delta$-Influence traces abnormal behavior back to the key poisoned training data utilizing one single affected test point, without assuming any prior knowledge of the attack. By retraining without these identified points, $\Delta$-Influence successfully unlearns multiple poisoning attacks across diverse datasets. We evaluate our method against five detection algorithms and apply five unlearning algorithms to the identified training set. Our results demonstrate that $\Delta$-Influence consistently outperforms existing approaches in all tested scenarios. Our findings highlight the potential of influence functions as a foundation for unlearning data poisoning attacks. Additionally, our ablation study sheds light on the strengths and limitations of various poisoning attacks and unlearning algorithms, offering insights that could inform the development of more effective unlearning techniques and robust poisoning attacks for rigorous testing.

## Acknowledgements

The authors would like to thank (in alphabetic order): Shashwat Goel, Shyamgopal Karthik, Elisa Nguyen, Shiven Sinha, Shashwat Singh, Matthias Tangemann, Vishaal Udandarao for their helpful feedback. AS acknowledges the Novo Nordisk Foundation for support via the Startup grant (NNF24OC0087820) and VILLUM FONDEN via the Young Investigator program (72069). WL, JL, and CSW acknowledges support from the Supervised Program for Alignment Research (SPAR) program. AP acknowledges financial support by the Federal Ministry of Education and Research (BMBF), FKZ: 16IS24085B and Open Philanthropy Foundation funded by the Good Ventures Foundation. We also thank the Center for AI Safety (CAIS) for their computational resources support.

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

# A    Connections to Existing Work

**Data Attribution: A Brief Overview**

The problem of training data attribution (TDA) has been explored using various approaches such as influence functions (Koh & Liang, 2017; Koh et al., 2019), Shapley value-based estimators (Ghorbani & Zou, 2019), empirical influence computation (Feldman & Zhang, 2020), and predictive datamodels (Park et al., 2023).

Broadly, TDA methods can be categorized into three groups: retraining-based methods, gradient-based methods, and predictive attribution models (see Hammoudeh & Lowd (2024) for a survey). Retraining-based methods systematically retrain models with and without specific training samples and observe changes in the model's outputs (Ghorbani & Zou, 2019; Jia et al., 2019; Feldman & Zhang, 2020). While these methods yield relatively accurate influence scores, they are computationally prohibitive for moderately large models, as the number of retrains often grows with the size of the training data. Gradient-based methods, such as influence functions (Cook & Weisberg, 1980), are computationally cheaper but often produce less reliable influence estimates for complex models (Basu et al., 2021).

Influence functions approximate the effect of individual training samples on a model's predictions by measuring how a prediction changes when a sample's weight is slightly perturbed. They were introduced to machine learning by Koh & Liang (2017) and have since been refined (Grosse et al., 2023; Kim et al., 2024; Pruthi et al., 2020). In data poisoning contexts, Seetharaman et al. (2022) used influence functions to mitigate degradation caused by previously identified poisoned data (Steinhardt et al., 2017). Building on this, we explore how advanced influence functions like EK-FAC (Grosse et al., 2023) can identify training examples disproportionately contributing to anomalous predictions in poisoned models.

Another approach, predictive data attribution, focuses on predicting model behavior directly based on training data (Ilyas et al., 2022; Park et al., 2023). While this approach can provide accurate influence estimates, the cost of training predictive models remains a significant limitation.

**Unlearning: A Brief Overview**

Machine unlearning, first proposed by Cao & Yang (2015), enables ML models to "forget" specific data points by removing their influence. This concept has gained importance with data protection regulations such as GDPR in the EU, which enforce the "right to be forgotten." Ideally, unlearning produces models equivalent to retraining from scratch after excluding the target data (Cao & Yang, 2015; Bourtoule et al., 2021; Gupta et al., 2021). However, retraining is computationally expensive, leading to the development of approximate unlearning methods (Ginart et al., 2019; Guo et al., 2020; Neel et al., 2021). These methods are often inspired by concepts from differential privacy, with the relevant $(\epsilon, \delta)$-provable unlearning definition formalized in Sekhari et al. (2021).

Recently, the scope of machine unlearning has expanded beyond privacy to address post-hoc system degradation, such as harmful knowledge removal (Li et al., 2024a) and adversarial attacks (Pawelczyk et al., 2024; Goel et al., 2024; Schoepf et al., 2024). In corrective unlearning, Pawelczyk et al. (2024) demonstrated the difficulty of mitigating strong poisons like Witches' Brew, while Goel et al. (2024) highlighted challenges when the complete set of manipulated data is unknown. These complexities underscore the inherent difficulty of the setting we address in this work.

**Data Poisoning Attacks** Data poisoning attacks are a significant threat to ML systems due to their ease of deployment and difficulty in detection. Even minor modifications to training data can lead to successful attacks on models trained on large datasets (Carlini et al., 2024). In this paper, we consider three forms of targeted data poisoning attacks: a *backdoor attack* (Gu et al., 2019) that adds a small patch in the corner of attacked images and modifies their labels to a target label, a *smooth trigger attack* (Zeng et al., 2021) that adds a trained pattern which is both hard to identify either in raw image domain or frequency domain, and *Witches' Brew* (Geiping et al., 2021), which adds a trained imperceptible pattern on attacked images without modifying labels. Note that the first two attacks modify the victim images' labels, while Witches' Brew is a *clean-label attack*. In this work, we do not consider indiscrimitive attacks which exhibit much weaker test-phase stealthiness as they degrade overall model performance, instead of forcing specific malicious predictions.

**Data Poisoning Defences** Defenses against data poisoning often involve trigger-pattern reverse engineering using clean data (Wang et al., 2019; Guo et al., 2019; Tao et al., 2022; Dong et al., 2021; Wang et al., 2020). These methods require additional steps such as input pre-filtering, neuron pruning, or fine-tuning (Liu et al., 2018; Chen et al., 2019; Li et al., 2021b; Zeng et al., 2022). Other approaches, like Anti-Backdoor Learning (Li et al., 2021a) and BaDLoss (Alex et al., 2024), necessitate tracking model updates and clean training samples, adding complexity to the defense process.

In contrast, our method requires access only to the trained model and a single poisoned test example without need to know any train poisons or attack patterns, offering a simpler yet effective defense mechanism.

## B   Experiment Details

### B.1   Poisoned Training Sample Is Not a Reliable Target for Influence-Based Unlearning

Given that a small subset of poisoned training data—commonly referred to as a forget set (Goel et al., 2024), could be identified as the prerequisite for unlearning, a natural question arises: why $\Delta-$ Influence focus on an identified affected test sample rather than simply using a poisoned training sample?

One overlooked fact is for some covert attack like Witches' Brew, the attack pattern is different between training and testing, which is not the case for Frequency Trigger and BadNet. Moreover, the clean-label attack manner and the imperceptible perturbations make it notoriously difficult to identify training poisons for such attacks. However, we emphasize that, regardless of how clever and stealthy an attack is designed, its primary goal is to alter model predictions on specific test points, making anomalies more apparent after deployment. Hence we think that having one identified test point is generally more feasible than identifying a poisoned training point in this context and better suited for influence-based analysis, which attributes model behavior to particular training instances.

What's more, as demonstrated in Table 9, taking Witches' Brew as an example, we find that even when defenders can reveal a poisoned training sample, the poisoned behavior cannot be reliably mitigated, while we show that $\Delta-$ Influence, utilizing an identified poisoned test point, can systematically undo the attack's impact.

This underscores a fundamental limitation of how influence functions work: influence functions inherently rely on clear causal relationships, where specific training samples directly impact corresponding test-time anomalies. However, in poisoned learning scenarios, such causality is often obscured: while the training poisons as a whole shifts model behavior, it's causal effect with one individual poisoned sample in it could be more ambiguous. This intuition that using a poisoned training sample as the target is less reliable than using an affected test point, is further supported by our empirical findings (as shown in table 9, influence-based methods fail to unlearn poisons when guided by a poisoned train point). We hope this observation provides useful insights for how target selection impacts causal tracing effectiveness in influence-based unlearning.

Based on the above observations, we suggest that using an identified poisoned test point for influence-based unlearning. Although when the attack pattern is consistent between training and testing (e.g. Frequency Trigger (Zeng et al., 2021) and BadNet (Gu et al., 2019)), using a poisoned training sample as the target also work, We argue that defenders should not assume such prior knowledge, e.g. what attack is performed and what the attack pattern is, which is rarely available in practice. Hence using a poisoned test point is more reliable and generalizable across different attack scenarios.

Finally, although it's not the focus of this paper, here we discuss approaches to get such a test point, realistic scenarios include: (i) whitehat adversarial research teams conducting jailbreaking-style tests to expose failure modes; (ii) Companies internally systematically stress-testing for vulnerabilities; and (iii) Companies using anomaly detection algorithms to monitor user interactions for abnormal behavior. Note that determining whether a test point is harmful or benign relies on the developer's domain expertise, this largely unexplored area is increasingly necessary due to massive training datasets and the rise of opaque open-source base models, offering promising directions for future research.

### B.2 Predefined Set for Image Augmentations

We employ a predefined set of standard image augmentation techniques: Flip, Rotation, Color Jitter, Elastic Transformation, Blur, Inversion, Color Switch, and Random Affine transform. For each transform, one augmentation is randomly selected from this set and applied to the affected test image.

### B.3 Attack Methods

The attack target and victim class are chosen at random for each trial. We shall now discuss the details for each attack method below. The relevant code is additionally publicly available in our repository.

**BadNet** For CIFAR datasets, we add a $3 \times 3$ checkboard-patterned black patch (pixel values set to zero) at the bottom-right corner of each $32 \times 32$ image. For the Imagenette dataset, we utilize a larger square $22 \times 22$ black patch to ensure successful injection of the poison. The number of poisoned images varies by dataset: 500 for CIFAR10, 350 for CIFAR100, and 858 for Imagenette.

**Smooth Trigger** The smooth trigger is generated for each dataset following the algorithm proposed in (Zeng et al., 2021). The number of poisoned images similarly varies by dataset: 500 for CIFAR10, 125 for CIFAR100 and 300 for Imagenette. Since the poison is more powerful, we are able to poison the model with less number of poisoned samples.

**Witches' Brew** The adversarial pattern is generated according to the method described in (Geiping et al., 2021). The number of poisoned images similarly varies by dataset: 500 for CIFAR10, 125 for CIFAR100 and 947 for ImageNette respectively. To ensure successful poisoning of Imagenette, we set we set eps=32, which is twice the value used for CIFAR10 and CIFAR100 (eps=16).

### B.4 Hyperparameters for Detection Methods

The hyperparameters are optimized through a grid search process to find the best possible values, following the process from Goel et al. (2024). Specifically:

**ActClust** We set the number of components, $n_{comp} = 3$, for all experiments. ActClust is quite robust a method, and we find that a value of 3 performs consistently best across all experiments.

**SpecSig** SpecSig involves two hyperparameters: the spectral threshold, used to identify significant singular values, and the contribution threshold, used to identify significant data point contributions. SpecSig is sensitive to both parameters. Typically, we select the best spectral threshold by grid search per dataset from the values 4, 6, 8, 10 and the contribution threshold from 7, 9, 11, 13. Higher values indicate a stricter constraint, resulting in fewer detected examples.

**FreqDef** For datasets with different image sizes, we train a specialized classifier following the methodology described in (Zeng et al., 2021).

**EK-FAC** We typically begin with a threshold value of 0 and select the best threshold among values (0, 10, 100, 500). Higher threshold values imply stricter filtering constraints, leading to fewer detected examples.

**TRAK** For this method, we evaluate a range of threshold values (0, 1, 2, 3, 4, 5) and choose the one yielding optimal results. In our experiments, a threshold of 0 frequently performs best. Higher threshold values imply stricter filtering constraints, leading to fewer detected examples.

**Ours** Similar to EK-FAC, starting with a threshold $\tau$ of 0 is generally effective where we search over (0, -1, -5, -10, -100). Lower threshold values and smaller $n_{tol}$ indicate stricter filtering constraints. For $n_{tol}$, We normally search over 0, 1, 2, 3, with 1 proving to be effective in most cases.

### B.5 Hyperparameters for SSD

Among the five unlearning methods considered, SSD is particularly sensitive to hyperparameters but is computationally efficient. This allows for lots of runs to select the optimal unlearning result. For each experiment, we evaluate all possible combinations of two SSD hyperparameters, the weight selection threshold,

which controls how protective the selection should be, and the weight dampening constant which defines the level of parameters protection. Specifically, we choose the weight selection threshold from values 2, 10, 50 and the weight dampening constant from 0.01, 0.1, 1.

## C   Results for Ablating Image-Only and Label-Only Augmentations

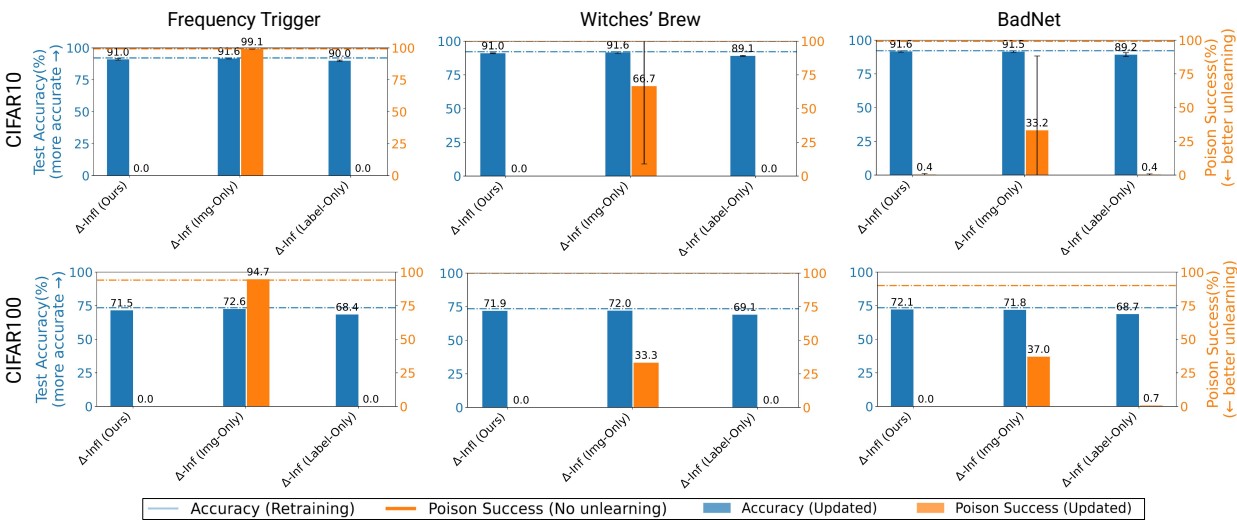

Figure 6: Poison Success Rate and Test Accuracy. This table shows both poison unlearning effectiveness and model utility. A method is considered successful if the poison success rate is below 5%. Label augmentations are instrumental towards identifying poisons, even in the clean-label poison cases. Figure structure from (Pawelczyk et al., 2024).

We show in Figure 6 that Label-Only augmentations are effective in removing the data poisoning (lower poison success rate), while Image-Only augmentations perform poorly in this regard. However, as demonstrated in Table 2, Label-Only augmentations lead to the unnecessary discard of many clean samples, whereas image augmentations significantly reduce the false positive rate, preserving clean data and improving detection precision. Therefore both label and image augmentations are crucial to the effectiveness of the Δ-Influence method.

## D   Scaling Experiments

### D.1   Will More Identified Poisoned Test Samples Improve Detection?

**Setup.** For attack methods such as Witches' Brew, only a single affected test point is identified. However, in cases where multiple test points can be identified, such as with BadNet Patch and Smooth Trigger attacks, we explore ways to enhance performance using two influence-based methods: $\Delta - \text{Influence}$ and EK-FAC, on the ImageNette dataset. Specifically, we show results when selecting five and ten test points to identify corresponding input points and determine their intersection as the poisoned data across both methods. This is done similarly to the $\Delta - \text{Influence}$ algorithm by retaining points with influence higher than the tolerance threshold, hence EK-FAC is additionally labeled (boosted).

**Results.** We showcase performance in Table 4 for BadNet poison and Table 5 for frequency trigger poison respectively. We observe a consistent trend: as the set of identified poisons increases, the precision improves significantly, leading to a substantial reduction in false positives and ultimately higher test accuracy.

**Conclusions.** Overall, identifying multiple poisoned test points enables more precise detection of poisons in the training set when using $\Delta - \text{Influence}$-like aggregation algorithms across test poisoned points. The

Table 4: **ImageNette BadNet.** For BadNet poison on the ImageNette dataset, increasing the number of identified test points significantly improves the precision. This enhancement leads to a notable reduction in false positives, thereby achieving higher overall test accuracy.

| Influence Methods | Precision($\uparrow$) | TPR($\uparrow$) | Poison Success Rate ($\downarrow$) | Test Accuracy ($\uparrow$) |
|---|---|---|---|---|
| **1 identified test point** | | | | |
| EK-FAC | 22.1% | 99.1% | 0.3% | 68.7% |
| $\Delta$-Influence | 49.0% | 100% | 0.8% | 79.7% |
| **5 identified test points** | | | | |
| EK-FAC | 25.9% | 98.8% | 0.5% | 73.3% |
| EK-FAC(boosted) | 34.2% | 98.5% | 0.8% | 75.4% |
| $\Delta$-Influence | 66.7% | 100% | 0.5% | 80.0% |
| **10 identified test points** | | | | |
| EK-FAC | 26.6% | 98.8% | 0.5% | 75.8% |
| EK-FAC(boosted) | 48.9% | 97.2% | 1.6% | 77.8% |
| $\Delta$-Influence | 67.2% | 100% | 0.8% | 79.9% |

Table 5: **ImageNette Frequency Trigger.** For frequency trigger poison on the ImageNette dataset, increasing the number of identified test points significantly improves the precision. This enhancement leads to a notable reduction in false positives, thereby achieving higher overall test accuracy.

| Influence Methods | Precision($\uparrow$) | TPR($\uparrow$) | Poison Success Rate ($\downarrow$) | Test Accuracy ($\uparrow$) |
|---|---|---|---|---|
| **1 identified test point** | | | | |
| EK-FAC | 10.5% | 99.3% | 0% | 72.4% |
| $\Delta$-Influence | 25.8% | 99.3% | 0% | 75.4% |
| **5 identified test points** | | | | |
| EK-FAC | 12.8% | 99.0% | 0% | 74.4% |
| EK-FAC(boosted) | 21.8% | 99.0% | 0.3% | 74.0% |
| $\Delta$-Influence | 27.5% | 99.3% | 0.3% | 76.6% |
| **10 identified test points** | | | | |
| EK-FAC | 12.9% | 99.3% | 0% | 74.1% |
| EK-FAC(boosted) | 24.2% | 99.3% | 0.3% | 73.6% |
| $\Delta$-Influence | 28.7% | 99.3% | 0.3% | 75.0% |

number of such test points that can be identified in practice often depends on the specific deployment scenario.

### D.2 Hyperparameters for ImageNette

To accommodate Imagenette's larger image sizes and increased complexity, we increase the patch size for BadNet poisoning, use a more intense trigger pattern for frequency-based poisoning, and poison a greater fraction of training images (10%). Additionally, for the Witches' Brew method, we relax the perturbation constraint, setting $\epsilon = 32$ instead of $\epsilon = 16$.

## E Does $\Delta$-Influence Perform the Best Across Unlearning Algorithms?

**Setup.** The probe was conducted across various detection methods; however, instead of employing the exact unlearning algorithm, we use a popular alternative algorithm called SCRUB which involves gradient ascent. We similarly measure the performance as well as the success rate of the poison removal were evaluated. Note the TPR rate and precision do not change.

**Results.** The evaluation results in Figure 7 shows that $\Delta-$Influence outperforms other methods, unlearning poisons in all six cases with minimal performance loss. In contrast, EK-FAC, ActClust, and SpecSig performed randomly, achieving unlearning primarily because even a randomly initialized model would not retain poisoning. Performance drops were primarily due to SCRUB's sensitivity to false positives from its gradient

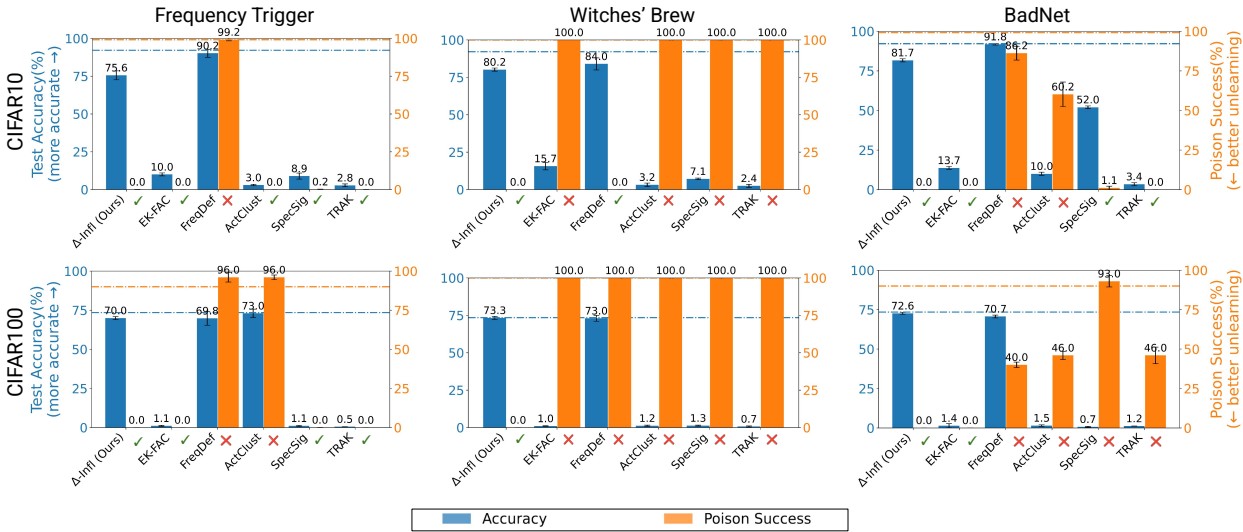

Figure 7: Poison Success Rate and Test Accuracy. with SCRUB Unlearning algorithm. This table shows both poison unlearning effectiveness and model utility. A method is considered successful if the poison success rate is below 5%, marked by ✓, with unsuccessful methods marked by ✗. Δ-Influence is successful in 6/6 cases, while the rest fail by not be distinguishable from a randomly initialized model. In contrast, Δ-Influence has only minor drops in test accuracy. Figure structure from (Pawelczyk et al., 2024).

ascent step. FreqDef avoided randomness but failed to unlearn poisons in all cases. Notably, Δ − Influence minimized false positives, maintaining consistent and reliable outcomes.

**Conclusions.** Δ−Influence proves to be remarkably robust even across unlearning methods which are highly sensitive to false positives. It achieves a 6/6 poison removal rate while incurring only minor performance losses due to false positives.

# F   Comparing Prior Knowledge Requirements Across Detection Methods

While Δ − Influence requires no prior knowledge about the attack implementation, several baseline methods rely, to varying degrees, on certain assumptions about the attack characteristics. We summarize these assumptions and their limitations below:

1. Frequency-Based Detection (FreqDef) implicitly assumes that poisoning attacks introduce modifications that manifest as detectable artifacts in the frequency domain.

**Limitation.** This assumption may not hold for attacks specifically designed to evade frequency analysis. In our experiments, the Frequency Trigger attack is deliberately crafted to be imperceptible in both spatial and frequency domains, resulting in FreqDef achieving very low TPR (3.2% on CIFAR10, 2.4% on CIFAR100, as shown in Table 1) and high PSR (99.3% on CIFAR10, 93.0% on CIFAR100, as shown in the first column of Figure 3).

2. Activation Clustering (ActClust) relies on two strong assumptions: (i) Poisoned samples form a separable cluster in the activation space (ii) This cluster is typically smaller than the clean cluster (*i.e.*, poisoning budget is limited).

**Limitation.** The method will naturally misidentify clean samples as poisons if an attacker increases the poisoning budget such that poisoned samples become the majority. Furthermore, for some attacks, poisoned samples might have activation patterns similar to clean data, violating the clustering assumption, which may partially explain why the Witches' Brew attack is hard to detect by ActClust (as shown in the second column of Figure 3).

3. Spectral Signature (SpecSig) assumes that, in the activation space, a backdoor trigger tends to leave behind a detectable trace in the spectrum of the covariance of a feature representation learned by the model.

**Limitation.** The effectiveness of SpecSig depends on whether the attack trigger indeed produces such spectral patterns . As shown in the first column of Figure 3, the Frequency Trigger attack is such a method hard to be detected by SpecSig.

## G  The Precision Challenge in Poisoning Detection

Despite the poisoning removal effectiveness achieved by $\Delta -$ Influence across various datasets and attack methods, the precision of all methods still remains relatively low. This reflects a fundamental challenge of poisoning detection: clean samples can exhibit characteristics similar to poisoned samples. This is not only true for the baseline methods, but also for influence-based methods: we think clean samples can also have high influence scores due to (i) the interconnected nature of learning, which means both that some clean samples might indeed help the model understand an abnormal prediction and even though the poison successfully changes the prediction, other clean signals and features, learned from clean data are still present and contribute to the forward pass of the model (ii) the approximation error of influence functions. Thus, high influence alone does not necessarily indicate poisoning and that's why we propose $\Delta -$ Influence to better separate poisoned samples from clean samples based on *Influence Collapse*.

To further improve precision, we suggest two possible directions:

Aggregated Filtering Mechanism. In addition to $\Delta -$ Influence, apply more filtering mechanisms to "purify" the detected set. For instance, use a clean validation set as an additional attribution target of influence functions. Samples flagged by $\Delta -$ Influence but showing high positive influence on clean predictions may likely be false positives and could be filtered out. Other non-influence-based filtering mechanisms are also worth considering.

Monitor Influence Change Across Different Attribution Targets. Similar to $\Delta -$ Influence, rather than using static influence scores, tracking how influence changes across different and carefully designed attribution targets might be worth paying attention to for future research. We believe this shift from "what is the influence?" to "how does influence change?" provides a more informative signal. We hope this methodology will inspire better influence-based detection work in the future.

## H  Robustness to Varied Trigger Patterns

We now provide additional experiments to evaluate our method's robustness when the defender's identified test point contains a trigger different from those in the poisoned training set. Specifically, for the BadNet attack on CIFAR-10, we tested two scenarios:

Normal Case. The test point trigger matches the training data triggers (both located in the bottom-right corner)

Varied Trigger Case. The training data triggers are in the bottom-right corner, while the test point trigger is in the upper-left corner

Table 6: **Robustness to Varied Trigger Patterns.** Performance of $\Delta -$ Influence when the test trigger location varies. We report (Test Accuracy / Poison Success Rate) for unlearning performance.

| | #Suspicious | Precision | TPR | EU | CF | SSD | Scrub | BadT |
|---|---|---|---|---|---|---|---|---|
| Normal | 2815 | 17.6% | 99.1% | 91.8% / 0.0% | 93.0% / 0.0% | 17.4% / 0.1% | 81.7% / 0.0% | 91.2% / 68.8% |
| Varied Trigger | 1978 | 24.6% | 97.2% | 92.4% / 0.0% | 93.0% / 0.0% | 12.0% / 0.0% | 91.2% / 0.0% | 91.9% / 26.8% |

As shown in Table 6, Delta-Influence demonstrates strong robustness in both scenarios and most unlearning methods (EU, CF, Scrub) successfully remove the poisoning effect in both cases, with test accuracy well maintained. These results align with our findings in Section 4 of the main paper.

## I Controlled Detection Budget Analysis

Table 7: **Controlled Detection Budget.** Comparison of detection and unlearning performance when controlling the number of suspicious samples to be approximately equal (∼2800).

| Method | #Suspicious | Precision | TPR | Test Acc / PSR |
|---|---|---|---|---|
| $\Delta$ − Influence (Ours) | 2815 | 17.6% | 99.1% | 91.8% / 0.0% |
| EK-FAC | 3048 | 0.3% | 1.8% | 91.0% / 99.7% |
| TRAK | 3172 | 0.06% | 0.4% | 91.2% / 99.7% |
| FreqDef | 2958 | 12.0% | 71.0% | 91.4% / 78.3% |
| SpecSig | 2563 | 13.2% | 67.8% | 91.9% / 86.6% |

Table 8: **Number of Suspicious Samples.** We provide the detailed number of suspicious samples identified by each method reported in Table 1 of the main paper.

| Method | CIFAR10 | | | CIFAR100 | | |
|---|---|---|---|---|---|---|
| | Frequency Trigger | Witches' Brew | BadNet | Frequency Trigger | Witches' Brew | BadNet |
| SpecSig | 33962 | 34571 | 12264 | 19600 | 14667 | 22239 |
| ActClust | 22523 | 22238 | 21568 | 20833 | 23000 | 21066 |
| FreqDef | 4000 | 4588 | 4519 | 3000 | 5444 | 5659 |
| TRAK | 25214 | 24901 | 24632 | 24200 | 30000 | 24999 |
| EK-FAC | 17241 | 10875 | 11982 | 13444 | 14750 | 7656 |
| $\Delta$ − Influence (Ours) | 3759 | 2939 | 2815 | 4310 | 3714 | 909 |

In addition to the results presented in Table 1 of the main paper, we report the exact number of suspicious samples identified by each detection method in Table 8. As we can see, different methods identify vastly different numbers of suspicious samples. To further analyze the detection performance under a controlled setting, we conduct additional experiments where we fix the number of detected suspicious points to be approximately equal across all methods by manually tuning their hyper-parameters. As shown in Table 7, for BadNet on CIFAR-10, we controlled the number of detected suspicious samples (#Suspicious) to approximately 2,800 across all detection methods, then applied the Exact Unlearning (EU) method for unlearning. Note that we do not include Activation Clustering (ActClust) in this controlled experiment because its clustering-based approach does not allow such fine-grained control. The results demonstrate that even when all methods flag a similar number of suspicious samples, $\Delta$ − Influence significantly outperforms baselines in both precision (17.6%) and TPR (99.1%), achieving complete poison removal (PSR = 0.0%) while maintaining high test accuracy (91.8%). In contrast, EK-FAC and TRAK exhibit extremely low precision (0.3% and 0.06%) and fail to identify true poisons (PSR > 99%), while FreqDef and SpecSig achieve moderate performance but still cannot adequately remove poisoning (PSR = 78.3% and 86.6%). This controlled comparison further validate $\Delta$ − Influence's superior performance.

## J Compute Resources and Time Cost Analysis

We used a single NVIDIA A100-80GB GPU for all experiments. Note that $\Delta$ − Influence has higher computational cost compared to non-influence-based baseline methods (e.g. Activation Clustering) because the influence function computation itself is time-costing. In our practice, on CIFAR-10 with $n_b = 50$ transformations, detection averages 33 minutes per affected test point (39.6 seconds per transformation).

We believe such computational cost is acceptable for corrective unlearning scenarios for several reasons. First, detection is a one-time cost triggered only when anomalies are observed. Second, the time can be *substantially* reduced through engineering optimizations: (i) parallelizing influence computations across transformations (which are independent), and (ii) leveraging ongoing improvements in influence function implementations. Third, and most importantly, preventing deployment of poisoned models justifies the computational invest-

ment—33 minutes for detection is negligible compared to the potential damage of undetected backdoors in safety-critical systems. We emphasize that our focus is on detection effectiveness rather than computational optimization and believe that the trade-off between computational cost and detection accuracy strongly favors accuracy in security-critical scenarios.

According to Grosse et al. (2023), the per-query time complexity of EK-FAC is:

$$\sum_{\ell=1}^{L} O(M_\ell^2 P_\ell + M_\ell P_\ell^2),$$

where $L$ is the number of layers, $M_\ell$ is the input dimension, and $P_\ell$ is the output dimension of layer $\ell$.

The computational cost of TRAK is approximately:

$$M \times (\text{model\_training\_time} + \text{gradient\_computation\_time}),$$

where $M$ denotes the number of trained models.

Additionally, the time complexities of ActClust and SpecSig are $O(Nk^2d)$ and $O(Nd^2)$ respectively. Here, $N$ represents the number of training samples, $d$ is the dimension of the target layer, and $k$ is the reduced dimension. For the FreqDef, the computational overhead is negligible, depending primarily on inference time once the detector model is trained.

We kindly refer readers to the original papers for more detailed analyses of the computational complexities of these baseline methods.

## K  The Causal Pitfall of Targeting a Poisoned Training Sample

**Setup.** We investigate whether directly using a known poisoned training sample as the attribution target for influence functions can still effectively detect and unlearn poisons (although such availability can be hard to achieve for attacks like Witches' Brew).

**Results.** As shown in Table 9, for CIFAR-10, when the attribution target is a training poison, $\Delta-$Influence fails to achieve successful unlearning. This is indicated by the Poison Success Rate remaining at 100%, which signifies that the attack remains fully effective. This inconsistent performance demonstrates that poisoned training samples are unreliable attribution targets for influence-based detection.

Table 9: **Failed Unlearning When Targeting a Known Poisoned Training Point.** Comparison of using an affected Test point versus a known Train poison as the attribution target.

| Identified Point | TPR($\uparrow$) | Poison Success Rate ($\downarrow$) | Test Accuracy ($\uparrow$) |
|---|---|---|---|
| | | **CIFAR10** | |
| Test | 19.4% | 0% | 91.0% |
| Train | 8.4% | 100% | 90.3% |
| | | **CIFAR100** | |
| Test | 62.4% | 0% | 71.9% |
| Train | 84.0% | 0% | 71.9% |

**Conclusion.** These results justify our choice of using the attribution target from the deployment phase (*i.e.*, test point) instead of training phase. The latter approach yields inconsistent unlearning performance because the causal dependency among poison peers is weaker than the collective influence of all poisons on an affected test point. Our choice is therefore grounded in a causal perspective: the goal is to find training examples responsible for a specific erroneous prediction, making the prediction itself the logical starting point.

## L    Limitations

$\Delta$ − Influence is based on influence functions and hence inherit their drawbacks. Possible attacks like those are only injected during test phase can evade our detection. Also, specially designed adversarial attacks against influence functions can hinder the effectiveness of our method. However, to our best knowledge, there are currently no poisoning attacks specifically designed to evade influence function-based detection. We hypothesize this may be because such an attack would face a fundamental tension: (1) To successfully cause abnormal prediction (the attack's objective), poisoned samples must significantly contribute to the affected test point's prediction no matter how the attack method is designed. Hence they, naturally, have high influence scores. (2) However, to evade influence-based detection, they would need to maintain low influence scores on that same test point. Reconciling these causal contradictory requirements might be inherently challenging. Nevertheless, we believe investigating robustness against potential adaptive attacks that might attempt to manipulate influence scores remains an interesting and important direction for future research.

