# OpenReview forum: "Delta-Influence: Identifying Poisons via Influence Functions"
_TMLR — Accepted by TMLR_

### Review · Reviewer_2vCv · 2025-11-11

**Summary Of Contributions:**

**Summary**: This paper studies the detection of targeted data poisoning attacks and backdoor attacks. Given a test sample affected by poisoning, the authors propose using a modified influence function to identify the corresponding poisoned samples in the training set. Specifically, they introduce the delta-influence function, which measures the difference in influence between a transformed test sample and its original version. The authors demonstrate the detection capability of their proposed method across three attacks and two datasets, comparing it against other detection methods and influence functions. They further remove the identified poisoned samples using existing unlearning methods.

**Key Strengths**: The paper addresses an important problem: detecting poisoned data under a unique threat model that assumes a given affected test sample. The problem is well-motivated by the gap in the unlearning literature, where identifying poisoned data is a prerequisite but remains understudied. The proposed delta-influence function is interesting and well-justified. The experiments convincingly demonstrate that the proposed method outperforms other detection methods.

**Key Weaknesses**: (1) The paper is not truly about unlearning; (2) The threat model is not well-defined and does not appear particularly practical; (3) The paper is restricted to targeted attacks only; (4) The experimental setting is vague and several takeaways are unclear.
I will elaborate on these points in detail below.

**Audience:**

Yes

**Audience Explanation:**

The paper would be of interest to audiences interested in detecting targeted data poisoning attacks and backdoor attacks.

**Claims And Evidence:**

No

**Claims Explanation:**

**[1. Data poisoning attacks]**: The paper studies a small subset of data poisoning attacks (one targeted attack and two backdoor attacks) and overlooks many other data poisoning attacks in the literature. Specifically, the paper assumes that each attack affects only one test sample (or those containing a trigger), which is not applicable to indiscriminate attacks that affect the entire test set. The authors should clearly identify the attack vectors studied and discuss the potential generalization of their observations to other attacks, including those effective against generative models.

**[2. Threat model]**: The threat model and assumptions underlying the detection process are not formally defined. The authors assume that the defender has access to a small set of affected test points, but how would one determine whether a test sample is affected by data poisoning or simply reflects a model deficiency? Moreover, targeted attacks and backdoor attacks present different challenges. For targeted attacks, does the defender know the true label? For backdoor attacks, does the defender know the trigger?

**[3. Necessity of the proposed method]**: Following the previous point, if a defender already possesses such knowledge about the test set, why would the proposed pipeline (influence function, detection, unlearning) be necessary? Since these attacks do not significantly affect model accuracy, simply correcting the misclassifications or removing the identified triggers would address the problem.

**[4. Unlearning]**: The title of the paper, "Unlearning Poisons via Influence Functions," suggests a new unlearning algorithm using influence functions. However, the paper is fundamentally about detecting poisons using influence functions, with unlearning as a natural next step (though whether this step is necessary still requires discussion). I believe the current presentation, especially the title, is misleading and needs to be revised.

**[5. Experimental Details]**:
- It is unclear how the authors select the poisoned test samples and determine the number of samples to consider. This raises the question of whether the results are biased. Specifically, the results in Figure 4 appear very different from the observations in Pawelczyk et al., and I wonder whether the authors chose the same test samples.
- The caption for Table 1 discusses PSR, which does not appear in the table. This is very confusing.
- The attribution analysis of the Witches' Brew poisoning subset is interesting, but it would benefit from generalization to a larger number of test samples.
- The authors should also include the time complexity of the proposed method compared to other baseline methods to justify its feasibility.

**Requested Changes:**

1. Include a literature review on data poisoning attacks, specify the studied types of attacks, and justify the choice.

2. Define the threat model and assumptions properly, and justify why the proposed method is necessary.

3. Adjust the presentation and titles, tune down the claim on "unlearning with influence function".

4. Add descriptions on how the test samples are selected and how many trials of experiments are performed (e.g., error bars).

5. Revise the caption of Table 1.

6. Specify the setting of Table 3.

7. Include time complexity to justify feasibility.

---

### Review · Reviewer_zZff · 2025-11-24

**Summary Of Contributions:**

This paper presents a defense against poisoning attacks that combines data attribution, an explainability method, with machine unlearning to identify and remove the poisoned training instances that negatively influence a prediction on a given test instance. In particular, the proposed method, called $\Delta$-influence, consists of three steps, consists of three steps, given only one suspicious test instance: for each training sample, apply the influence function method, manipulating the test instance through label flipping and a set of input transformations (e.g., rotation), thereby creating an ensemble of influence scores, one for each transformation; (ii) if more than a certain fraction of these scores are negative, classify the corresponding training instance as poisoned; (iii) use an unlearning method to remove the flagged poisoned instances.

The experimental evaluation considers two computer vision datasets, five defenses against poisoning and three poisoning attacks from prior work. The evaluation is comprehensive and compares the proposed defense against existing methods, it assesses its performance under different attacks and includes ablation studies to quantify the impact of the input transformations and the effect of unlearning only the detected poisoned instances on model accuracy and robustness. The results highlight that the proposed method is more effective at identifying the correct poisoned instances than other defenses, although the overall efficacy of the framework also depends on the chosen unlearning algorithm.

**Additional Comments:**

I invite the authors to address the following typos and editorial issues:
- Please check the use of parentheses in:
   - _Pawelczyk et al. (2024)_, page 1.
   - _(Chundawat et al., 2023))_, page 5.
- On page 1, the sentence “_To address this limitation, we introduce ∆−Influence, a novel approach that enhances influence_” appears to present the method for the first time, although ∆−Influence has already been introduced earlier in the text. I would suggest rephrasing this sentence here.
- The domain of the pair $z = (x, y)$ in Section 2.1 should be defined.

**Audience:**

Yes

**Audience Explanation:**

The scientific community is concerned about the effects of data poisoning and the potential ineffectiveness of unlearning against such attacks. A technique that can accurately identify and remove poisoned training data from just one compromised test instance and that outperform the other state-of-the-art methods would certainly be of interest to the scientific community that reads TMLR.

**Broader Impact Concerns:**

No impact concerns to signal.

**Claims And Evidence:**

Yes

**Claims Explanation:**

The experimental evaluation is comprehensive and effectively supports the effectiveness of $\Delta$-Influence compared to popular baselines for detecting and unlearning poisoned instances in the training set. The proposed method is more precise in detection, as demonstrated by its higher true positive rate and precision relative to the baselines, and it also has a more desirable effect on the affected classifier, since it effectively removes the poisoning impact without inducing a significant loss in accuracy. Moreover, the ablation study quantifies in a clear way the contribution of the different components of the method, such as the use of input transformations and the choice of unlearning algorithm.

**Requested Changes:**

I would suggest the following additions to the paper, which are desiderata rather than conditions for acceptance:
- In Section 3.2, in the “Conclusion” paragraph, it is stated that the proposed method requires no prior knowledge about the poisoning attack. This is correct, but it would be even better to clarify, for comparison purposes, which of the other baselines do require knowledge about the attack. Such a clarification would further highlight the merits of the proposed method.
- It would be helpful to provide some intuition regarding the results in Table 1. Why is the true positive rate of all baselines, not only the proposed method, below 50%? Can we derive any insights that might guide future improvements in detection methods? While the proposed method obtains a higher true positive rate than the baselines, providing an explanation for this point would be a nice addition to the paper.
- The paper states as a limitation that adversarial attacks against influence functions can hinder the effectiveness of the proposed method (Appendix F). Demonstrating the robustness of the method against some of these attacks could further strengthen the work, although this is not necessary and could be left for future investigation (another paper ;) ).

---

### Review · Reviewer_ZEF4 · 2025-12-12

**Summary Of Contributions:**

This paper introduces a method for identifying compromised training samples based on the phenomenon of influence collapse. It proposes a Trojan behavior mitigation algorithm that unlearns identified corrupted samples. Specifically, the paper suggests measuring changes in influence function values after applying multiple transformations to a single captured Trojan example. The authors observed that for Trojan-corrupted training samples, the influence function value significantly decreases after transformation, unlike in clean examples. This phenomenon is termed influence collapse. The proposed algorithm identifies Trojaned training data points by examining the proportion of predefined transformations that exhibit influence collapse. Finally, the paper proposes eradicating Trojan behavior by unlearning the identified suspicious data points.

Empirical studies were conducted using three publicly available datasets under three different backdoor attack scenarios. Multiple detection baseline methods were compared to demonstrate the advantages of the proposed method. Additionally, ablation studies were performed to further support the contributions of the research.

**Audience:**

Yes

**Audience Explanation:**

The paper addresses AI security and the mitigation of backdoor attacks on neural networks, a critical topic in contemporary AI research. This subject is likely to be of significant interest to the TMLR audience.

**Broader Impact Concerns:**

Not relevant

**Claims And Evidence:**

No

**Claims Explanation:**

To thoroughly evaluate the contribution of the proposed method, I would appreciate further clarification on several points:

Trigger Variability in Attacks: It is common for attackers to employ multiple triggers rather than a fixed one. For instance, in the BadNet attack, triggers are typically placed randomly on the four corners of corrupted images with position jittering, rather than being fixed. How does the algorithm perform when the captured Trojan examples have triggers that differ from those used in the rest of the training set?

Impact of Filtering Threshold: It appears that precision, rather than recall or AUC, is the key driver of mitigation performance in fixing Trojaned network. This leads to my second question: What is the total number of data points identified by each baseline method? It seems that $\Delta$-Infl identifies fewer data points than other methods but with higher precision. What would be the outcome if all methods identify the same sets of corrupted data by adjusting the filtering threshold and then unlearned roughly the same number of suspicious data points, instead of selecting hyper-parameters by optimizing Precision/TPR/AUROC on a validation set?

Relevance of the Unlearning Algorithm: The unlearning algorithm appears somewhat peripheral to the main contribution of the study. As shown in Table-1, while $\Delta$-Infl exhibits superior precision, it underperforms in AUC compared to baseline methods across several settings, complicating the interpretation of results. It is important to consider that some mitigation algorithms might prioritize a high recall detector over a high precision one. A straightforward heuristic approach could involve fine-tuning the Trojaned network on the complementary set, rather than unlearning the detected set.

**Requested Changes:**

I would appreciate the inclusion of the following ablation study results to enhance the paper:

Detection of Corrupted Points with Varied Triggers: it would be valuable to conduct experiments on detecting corrupted points where the unique captured Trojan example has a trigger different from those used in the rest of the dataset. For instance, how does the algorithm perform if the defender's example has a trigger on the bottom right corner, while all other triggers in the corrupted database are located in the upper left corner? Similarly, how does the algorithm respond if the defender's example uses a trigger generated with low-pass filter 1, whereas others use low-pass filter 2?

Ablation on Filter Threshold: Please report the number of suspicious data points identified by each method. Instead of tuning the filtering threshold on a validation set, consider selecting a threshold to identify approximately the same number of corrupted data points across methods. Varying this predetermined number could provide better insights into the transition of performance.

Finetune on complementary set: It would be advantageous for the authors to investigate fine-tuning the Trojaned model on the complementary set, rather than solely concentrating on unlearning the detected set. Conducting this ablation study could help mitigate bias in selecting an unlearning algorithm that favors a high-precision detector over a high-recall detector. This analysis should be performed in comparison with the strongest competitors listed in the baseline.

---

### Decision · Action_Editor_JKHu · 2026-01-20

**Recommendation:** Accept with minor revision

**Additional Comments:**

The main contribution of this work is a simple differencing approach (based on influence functions) for detecting poisoned training samples. The authors were able to address most of the reviewers' concerns during the discussion phase and significantly improved the presentation and clarity of their draft. All reviewers leaned towards acceptance in the end.

I agree with Reviewer 2vCv on Figure 4/5: while the motivation to remove confounding issues seems reasonable, one must take into consideration that no one is incentivized to use an *approximate* unlearning method when spending the same computational budget as retraining. It is recommended to combine Figure 4 with Figure 8, where one makes it clear that EU and CF used 100% computational budget while SSD, Scrub and BadT used 10% computational budget (i.e., remove results for approximate unlearning methods with 100% computational budget). The danger with the current separate figures is that a casual reader might conclude from Figure 4 that approximate unlearning methods are doing pretty well on this task, blissfully unaware of the computational budget issue (which the authors did not seem to explicitly mention in the main paper). Same recommendation regarding Figure 5 and Figure 9.

**Audience:**

Yes

**Audience Explanation:**

The proposed method offers a simple way to detect poisoned samples, which could enhance the security of ML models deployed in practice and be of interest to many practitioners.

**Claims And Evidence:**

Yes

**Claims Explanation:**

The authors conducted extensive experiments (especially during the discussion phase) to support their proposed method and to address the reviewers' questions.